

# Genome-wide identification, classification and expression profile analysis of the HSF gene family in *Hypericum perforatum*

Li Zhou, Xiaoding Yu, Donghao Wang, Lin Li, Wen Zhou, Qian Zhang, Xinrui Wang, Sumin Ye and Zhezhi Wang

National Engineering Laboratory for Resource Development of Endangered Crude Drugs in Northwest China, Key Laboratory of the Ministry of Education for Medicinal Resources and Natural Pharmaceutical Chemistry, College of Life Sciences, Shaanxi Normal University, Xi'an, Shaanxi, China

## ABSTRACT

Heat shock transcription factors (HSFs) are critical regulators of plant responses to various abiotic and biotic stresses, including high temperature stress. HSFs are involved in regulating the expression of heat shock proteins (HSPs) by binding with heat stress elements (HSEs) to defend against high-temperature stress. The *H. perforatum* genome was recently fully sequenced; this provides a valuable resource for genetic and functional analysis. In this study, 23 putative *HpHSF* genes were identified and divided into three groups (A, B, and C) based on phylogeny and structural features. Gene structure and conserved motif analyses were performed on *HpHSFs* members; the DNA-binding domain (DBD), hydrophobic heptad repeat (HR-A/B), and exon-intron boundaries exhibited specific phylogenetic relationships. In addition, the presence of various *cis*-acting elements in the promoter regions of *HpHSFs* underscored their regulatory function in abiotic stress responses. RT-qPCR analyses showed that most *HpHSF* genes were expressed in response to heat conditions, suggesting that HpHSFs play potential roles in the heat stress resistance pathway. Our findings are advantageous for the analysis and research of the function of HpHSFs in high temperature stress tolerance in *H. perforatum*.

## INTRODUCTION

High temperature as an abiotic stress triggered by global warming is largely the result of deforestation and increases in atmospheric $CO_2$ concentrations. Global warming has caused worldwide declines in the yield of crops including wheat, rice, maize, and soybean, which are the most widely consumed staple foods in the world and feed over 50% of humanity (*Mittler, Finka & Goloubinoff, 2012*; *Sadok, Lopez & Smith, 2020*; *Song et al., 2018*; *Zhu et al., 2019*). High temperature has a pernicious impact on plants, such as oxidative stress and membrane permeabilization, due to effects on photosynthetic efficiency and decreased grain weight. Plants deploy several responses to mitigate high

Corresponding author
Zhezhi Wang, zzwang@snnu.edu.cn

temperature stress. The physiological and biochemical processes of stomatal development, shade avoidance response, antioxidant defense, and selective autophagy play important roles in adaptation to high temperature stress. These processes are regulated by essential genes and specific transcriptional factors that are involved in modulating mechanisms and the alleviation of high-temperature stress (*Samakovli et al., 2020*; *Thirumalaikumar et al., 2020*). Under heat stress, heat shock transcription factors (HSFs) can activate the rapid accumulation and expression of heat shock proteins (HSPs) to reduce heat-related damage such as electrolyte leakage, overproduction of reactive oxygen species (ROS) and oxidative damage (*Bokszczanin, 2013*). Many HSPs play critical roles in protecting plants from heat-induced damage as well as in protein folding, aggregation, degradation, and intracellular distribution (*Lin et al., 2011*; *Mittler, Finka & Goloubinoff, 2012*). In the heat shock reaction process, HSFs regulate the expression of heat stress-inducible genes by recognizing the heat stress elements (HSEs) present in the promoter regions of HSP genes (*Scharf et al., 2012*). Specifically, HSFs utilize their oligomerization domains to form trimmers and exert their functions as sequence-specific trimeric DNA-binding proteins. Previous studies have shown that transcription activation in vivo requires HSEs in HSF protein binding. The HSF recognizes a typical 5 bp sequence, 5-nGAAn-3, which forms at least three contiguous inverted repeats in the downstream HSP promoter (*Saha et al., 2019*). The highly conserved DNA-binding domain (DBD) in the N-terminus includes one three-helical bundle ($\alpha$1, $\alpha$2, $\alpha$3) and one antiparallel four-stranded $\beta$-sheet ($\beta$1, $\beta$2, $\beta$3, $\beta$4) to form a helix-turn-helix structure. The DBD domain is required for HSE specific binding to regulate the expression of downstream genes (*Guo et al., 2016*). The oligomerization domain (OD), also known as the HR-A/B region, has the characteristics of a coiled-coil structure and plays a role in transcription factor activity. It is mainly located at the C-terminal end of the HSF and connected to the DBD through a flexible linker comprising a heptad pattern of hydrophobic amino acid residues (*Lin et al., 2015*). A nuclear localization signal (NLS) is also present at the C-terminal of the HR-A/B region, consisting of a cluster of basic amino acids rich in lysine and arginine residues, and is essential for nuclear import; some HSF genes also have a nuclear export signal (NES) in the C-terminus, which contains several leucine residues and is crucial for regulating the nucleocytoplasmic distribution of HSF proteins (*Chidambaranathan et al., 2018*). Some HSF proteins also have short peptide motifs (AHA motifs) close to the C-terminal for transcriptional activator functions (*Kotak et al., 2004*). Based on analysis of the conserved DBD domain and HR-A/B regions, HSFs in plants are classified into three main classes (class A, B, and C) (*Nover et al., 2001*). The number of amino acid residues connecting DBD to HR-A/B differs among the three subgroups: Classes A, B, and C contain 9–39, 50–78, and 4–49 amino acid residues, respectively (*Miller & Mittler, 2006*; *Prandl et al., 1998*). Moreover, the number of amino acids linking HR-A and HR-B also varies considerably in different subgroups. There are 21 and seven amino acid residues inserted into the HR-A/B region in class A and class C, respectively, whereas this region in class B HSFs is compact, without insert sequences, between the heptad repeats (*Baniwal et al., 2004*). Additionally, the AHA motifs, which function by binding transcription protein complexes to activate the transcription of HSPs, are unique to class A members (*Scharf et al., 2012*). Recently,

HSF gene families were analyzed in different species, including maize, rice, pepper, tomato, soybean, and flax. Genome-wide analysis indicated that HSF proteins in various species may have different functions in reducing damage to high-temperature stress and also provide resources for evolutionary analysis (*Guo et al., 2015*; *Saha et al., 2019*; *Yang et al., 2016*).

*Hypericum perforatum* is an herbaceous perennial plant in the family Hypericaceae, the well-characterized secondary metabolites and pharmacological activities of which have attracted the attention of researchers (*Galeotti, 2017*). Substances present in the extracts of *H. perforatum* include acyl-phloroglucinols, naphthodianthrones, xanthones, and flavonoids, and these pharmacological compounds have been demonstrated to have antiviral, antitumor, antiinflammatory, antimicrobial, and antioxidant effects (*Nahrstedt & Butterweck, 2010*). However, *H. perforatum* production and quality are challenged by various environmental stresses, such as cold, high temperature, and drought (*Lausen, Emilsson & Jensen, 2020*; *Skyba et al., 2012*; *Zobayed, Afreen & Kozai, 2005*). Therefore, it is important to characterize the stress-resistance genes of *H. perforatum*. In the current study, we identified 23 *HpHSF* genes and analyzed their physical and chemical characteristics, conserved domains, gene structures, evolutionary relationships, and *cis*-acting elements. Moreover, we explored expression profiles across four different tissues. In conclusion, this study provides a foundation for improved exploration of *HpHSF* gene function in *H. perforatum*.

## MATERIALS & METHODS

### Plant materials and treatment

Seeds of *H. perforatum* preserved in our laboratory were germinated and grown on a seedling bed in a greenhouse ($25 \pm 2$ °C, natural lighting). Humidity was maintained at 60%–80%. Two-month-old *H. perforatum* seedlings were transferred to an incubator maintained at 42 °C for heat stress treatments, and five time points (0, 1, 3, 6, and 12 h) were selected for sample collection. The entire seedling was collected for expression analysis of *HpHSF* genes under heat stress treatment. In addition, different tissue samples, including flowers, leaves, stems, and roots, were taken from two-year-old plants. All samples were collected in three replicates, and the samples were immediately immersed in liquid nitrogen and stored at $-80$ °C for RNA isolation.

### Identification of HpHSF members

The whole genome sequences of the HSF proteins in *H. perforatum* were detected and assembled by our laboratory (*Zhou et al., 2020a*; *Zhou et al., 2020b*). For HSF identification, the conserved amino acid sequence of a DNA-binding domain (Pfam: PF00447) was used to search the *H. perforatum* genome. Moreover, the HSF sequences of *Capsicum annuum* L. (pepper), *Vitis vinifera* L. (grape), and *Arabidopsis thaliana* obtained from plantTFDB (http://planttfdb.gao-lab.org/) were used as BLAST queries against the *H. perforatum* genome. All output genes with default were searched for conserved DNA-binding domains using Interpro (http://www.ebi.ac.uk/interpro/) and SMART (http://smart.embl-heidelberg.de/). In addition, the candidate genes were analyzed using

MARCOIl (http://toolkit.tuebingen.mpg.de/marcoil) to retain genes with a coiled-coil structure. The detected genes are listed in Table S1.

## Phylogenetic relationship analysis and sequence analysis

Full-length amino acid sequences of HSF from *A. thaliana*, *C. annuum* L., *V. vinifera* L. (grape), and *H. perforatum* were aligned using Clustal X; the extension penalty and opening penalty of gap were 0.2 and 10, respectively. The cut-off for delay divergent sequences was set to 40%. Residue-specific and hydrophilic penalties were applied in alignment. Then, the phylogenetic tree was inferred using MEGA 6.0. The statistical method used was the neighbor-joining (NJ) method, and the test of phylogeny was based on the bootstrap method with 1000 bootstrap replicates and pairwise deletion. The amino acid substitution model used was the p-distance model. Parameters including molecular weight, isoelectric point, aliphatic index, instability index, percentage of negatively/positively charged residues, and GRAVY of HpHSF proteins were displayed using the ExPASy database (https://www.expasy.org/). Furthermore, conserved motifs in HpHSF proteins were searched using Heatster (Heatster, https://applbio.biologie.uni-frankfurt.de/hsf/heatster/) and the exon/intron organization of HpHSF proteins was obtained using the Gene Structure Display Server program (GSDS, http://gsds.cbi.pku.edu.cn/). The *cis*-acting elements of 1.5 kb upstream sequences of the transcription initiation site in the promoter region of *HpHSF* genes were analyzed using PlantCARE (http://bioinformatics.psb.ugent.be/webtools/plantcare/html/). SWISS-MODEL server (https://swissmodel.expasy.org) programs were used to build and generate the three-dimensional structures of the HSF proteins.

## Isolation of RNA and cDNA synthesis

Total RNA from *H. perforatum* samples were isolated using the HiPure Total RNA Mini Kit following the manufacturer's protocol (Magen, China). The concentration of the isolated total RNA was determined using a NanoDrop 2000c spectrophotometer (Thermo Scientific, USA), and the integrity of the RNA was directly quantified by running agarose gel (1% w/v) at 150 V for 10 minutes. One microgram of RNA was used for first strand cDNA synthesis using PrimeScriptTM RT Reagent Kit (TaKaRa, China) according to the manufacturer's instructions.

## Primer design and qRT-PCR analysis

The primers for the 23 *HpHSF* genes were designed by GenScript (https://www.genscript.com). The parameters were: PCR amplicon size range: 100–180; primer Tm: minimum, optimum, and maximum: 59.5 °C, 60 °C, 60.5 °C, respectively; probe Tm: minimum, optimum, and maximum: 62 °C, 66 °C, and 70 °C, respectively. The specificity of the primers was determined using Bioedit by searching the primers given by GenScript against the *H. perforatum* genome (Table S2). In addition, qRT-PCR was performed on the LightCycler 96 system (Roche Diagnostics GmbH) using the ChamQTM SYBR®qPCR Master Mix (Vazyme, Nanjing, China) following the manufacturer's instructions. *HpActin-2* was used as an internal control (*Zhou et al., 2019*). The final cycle threshold (Ct) values were the mean of three values for each sample and three technical replicates, and the

2- $\Delta$ $\Delta$Ct method was used to analyze the relative changes in gene expression (*Livak & Schmittgen, 2001*). Data were analyzed using one-way ANOVA in the GraphPad Prism software (*, $P < 0.05$; **, $P < 0.01$; ***, $P < 0.001$). qRT-PCR was performed with three biological replicates for each sample, and each sample consisted of three technical replicates. The primers for the *HpHSF* genes used for qRT-PCR analyses are listed in Table S1.

## RESULTS

### Identification and isolation of *HSF* genes in *H. perforatum*

Twenty-three genes were identified as members of the HSF transcription factor family in *H. perforatum* based on a conserved DBD domain search and coiled-coil structure detection. These genes were named 'HpHSF' with consecutive numbers. More detailed information about HpHSF01 to HpHSF23 is shown in Table 1. The identified *HpHSFs* encoded 188–501 amino acids (average 345 aa), and molecular weights (MW) ranged from 21.72 to 54.91 kDa (average 39.15 kDa). The isoelectric points (pI) of HpHSFs varied from 4.79 to 8.86. Of the 23 *HpHSF* genes, the percentages of negatively charged residues (ASP + Glu) (n.c.r.) and positively charged residues (Arg + Lys) (p.c.r.) were 11.0%–17.6% and 8.4%–15.8%, respectively. According to the instability index analysis, all the HpHSF proteins were found unstable. In addition, the aliphatic index (A.I.) ranged from 54.52 to 76.18, and the grand average of hydropathicity (GRAVY) ranged from −0.826 to −0.523.

### Conserved domains of *HpHSF*s

The DBD and HR-A/B conserved domains were observed in the all of the *HpHSF* genes to reveal the sequence of conserved regions between members of the HpHSFs; multiple alignments of 23 *HpHSFs* were obtained using DNAMAN. The DBD domain close to the N-terminal was highly conserved (Fig. 1). The secondary structure prediction showed that the majority of the DBD domains consist of a four-stranded antiparallel $\beta$-sheet and three α-helices (α1–α3). In addition, MARCOIL was used for predicting the coiled-coil structure, which is characteristic of the HR-A/B regions adjacent to the DBD domain in the C-terminal. The 23 candidate HpHSF protein sequences were all determined to have a coiled-coil structure. The multiple alignment results of the HR-A/B regions showed that the HpHSF protein family could be divided into three classes based on the insertion amino acid residues between the A and B parts of the HR-A/B motif (Fig. 2).

### Phylogenetic relationships among *HpHSF* genes

To investigate the evolutionary relationships of the *HpHSF* genes, a total of 88 *HSFs*, comprising 21 from *Arabidopsis*, 25 from pepper, 19 from grape, and 23 from *H. perforatum* were used for phylogenetic tree construction using MEGA6.0. HSFs were clearly classified into three main groups, namely HSF A, B, and C (Fig. 3). HpHSF A was the largest group, representing 52.2% of the total HpHSFs; the second was HpHSF B, which represented 39.1%; and HpHSF C was the smallest group, which represented 8.7%. In addition, HpHSF A is classified into 9 subgroups (A1-A9) and includes 12 members (*HpHSF07*, *HpHSF18*, *HpHSF12*, *HpHSF11*, *HpHSF16*, *HpHSF21*, *HpHSF17*, *HpHSF02*, *HpHSF23*, *HpHSF13*,

Zhou et al. (2021), *PeerJ*, DOI 10.7717/peerj.11345

**Table 1** **The HSF genes identifed from the *H. perforatum*.**

| Gene name | Transcript ID | Length | | | MW (kDa) | pI | n.c.r. (%) | p.c.r. (%) | I.I. | Stability | A.I. | GRAVY |
|---|---|---|---|---|---|---|---|---|---|---|---|---|
| | | Protein | CDS | Gene | | | | | | | | |
| *HpHSF01* | HperS113g0097 | 293 | 882 | 1058 | 32.23 | 5.05 | 43 (14.7%) | 35 (11.9%) | 57.40 | unstable | 75.26 | −0.523 |
| *HpHSF02* | HperS020g0043 | 381 | 1146 | 1687 | 43.75 | 5.51 | 59 (15.5%) | 48 (12.6%) | 59.80 | unstable | 71.55 | −0.752 |
| *HpHSF03* | HperS219g0006 | 327 | 984 | 1842 | 37.9 | 7.29 | 36 (11.0%) | 36 (11.0%) | 47.69 | unstable | 72.14 | −0.660 |
| *HpHSF04* | HperS024g0021 | 222 | 669 | 1798 | 25.95 | 7.72 | 34 (15.3%) | 35 (15.8%) | 52.96 | unstable | 73.24 | −0.796 |
| *HpHSF05* | HperS024g0048 | 196 | 591 | 2959 | 22.47 | 6.85 | 31 (15.8%) | 31 (15.8%) | 46.57 | unstable | 69.08 | −0.747 |
| *HpHSF06* | HperS245g0169 | 226 | 681 | 3135 | 25.88 | 6.86 | 34 (15.0%) | 34 (15.0%) | 48.26 | unstable | 69.38 | −0.737 |
| *HpHSF07* | HperS025g0041 | 434 | 1305 | 1764 | 48.47 | 5.22 | 64 (14.7%) | 47 (10.8%) | 58.65 | unstable | 76.18 | −0.577 |
| *HpHSF08* | HperS254g0338 | 376 | 1131 | 1300 | 42.04 | 5.67 | 45 (12.0%) | 39 (10.4%) | 66.18 | unstable | 64.84 | −0.655 |
| *HpHSF09* | HperS338g0001 | 330 | 993 | 1169 | 36.57 | 5.67 | 43 (13.0%) | 38 (11.5%) | 50.96 | unstable | 60.24 | −0.600 |
| *HpHSF10* | HperS346g0011 | 428 | 1287 | 1897 | 48.01 | 4.91 | 66 (15.4%) | 44 (10.3%) | 56.52 | unstable | 67.64 | −0.642 |
| *HpHSF11* | HperS346g0247 | 324 | 975 | 1054 | 37.51 | 5.91 | 45 (13.9%) | 38 (11.7%) | 57.46 | unstable | 59.85 | −0.813 |
| *HpHSF12* | HperS362g0014 | 409 | 1230 | 1464 | 46.16 | 5.02 | 65 (15.9%) | 43 (10.5%) | 57.83 | unstable | 65.99 | −0.745 |
| *HpHSF13* | HperS388g0082 | 403 | 1212 | 2262 | 46.49 | 4.79 | 71 (17.6%) | 46 (11.4%) | 47.52 | unstable | 65.56 | −0.764 |
| *HpHSF14* | HperS398g0019 | 195 | 588 | 1981 | 22.35 | 8.86 | 26 (13.3%) | 30 (15.4%) | 63.69 | unstable | 58.10 | −0.818 |
| *HpHSF15* | HperS042g0257 | 248 | 747 | 1956 | 27.78 | 5.78 | 40 (16.1%) | 37 (14.9%) | 46.02 | unstable | 61.33 | −0.817 |
| *HpHSF16* | HperS434g0151 | 501 | 1506 | 2644 | 54.91 | 4.87 | 64 (12.8%) | 42 (8.4%) | 59.15 | unstable | 67.60 | −0.608 |
| *HpHSF17* | HperS044g0424 | 483 | 1452 | 2149 | 53.92 | 4.99 | 63 (13.0%) | 42 (8.7%) | 51.71 | unstable | 74.66 | −0.533 |
| *HpHSF18* | HperS443g0073 | 397 | 1194 | 1521 | 44.95 | 4.94 | 63 (15.9%) | 41 (10.3%) | 62.46 | unstable | 70.48 | −0.723 |
| *HpHSF19* | HperS006g0172 | 188 | 567 | 1981 | 21.72 | 8.54 | 25 (13.3%) | 28 (14.9%) | 52.32 | unstable | 54.52 | −0.797 |
| *HpHSF20* | HperS064g0032 | 455 | 1368 | 1950 | 51.78 | 5.91 | 64 (14.1%) | 57 (12.5%) | 62.11 | unstable | 71.12 | −0.644 |
| *HpHSF21* | HperS068g0017 | 495 | 1488 | 2278 | 54.66 | 4.96 | 65 (13.1%) | 45 (9.1%) | 56.75 | unstable | 67.39 | −0.650 |
| *HpHSF22* | HperS079g0626 | 270 | 813 | 1208 | 31.63 | 6.33 | 33 (12.2%) | 28 (10.4%) | 47.61 | unstable | 64.59 | −0.826 |
| *HpHSF23* | HperS091g0277 | 363 | 1092 | 1604 | 41.62 | 5.39 | 57 (15.7%) | 43 (11.8%) | 61.47 | unstable | 69.53 | −0.784 |

**Notes.**

MW (kDa), Molecular weight in kilo Dalton; pI, isoelectric point; n.c.r, total number of negatively charged residues (Asp +Glu); p.c.r, total number of positively charged residues (Arg +Lys); I.I., instability index; A.I., aliphatic index; GRAVY, grand average of hydropathicity.

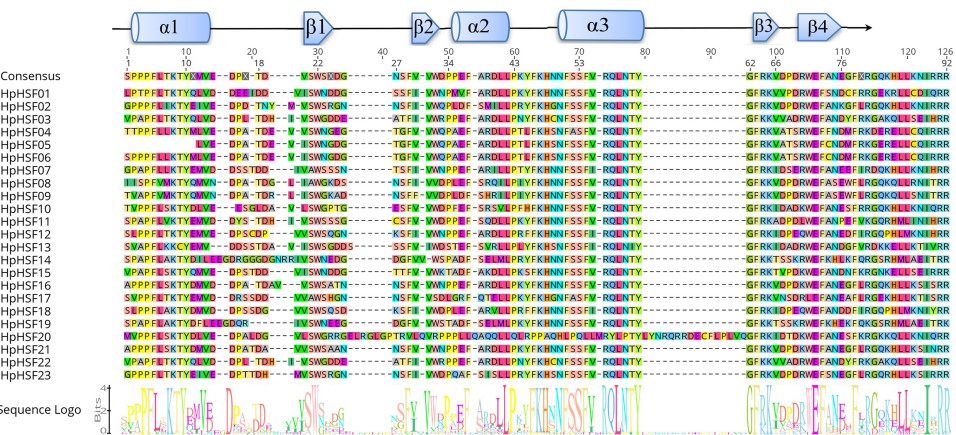

**Figure 1 Multiple sequence alignment of the DBD domains of 23 members of the HSF protein family.** Three α-helices and four β-sheets were presented in the region.

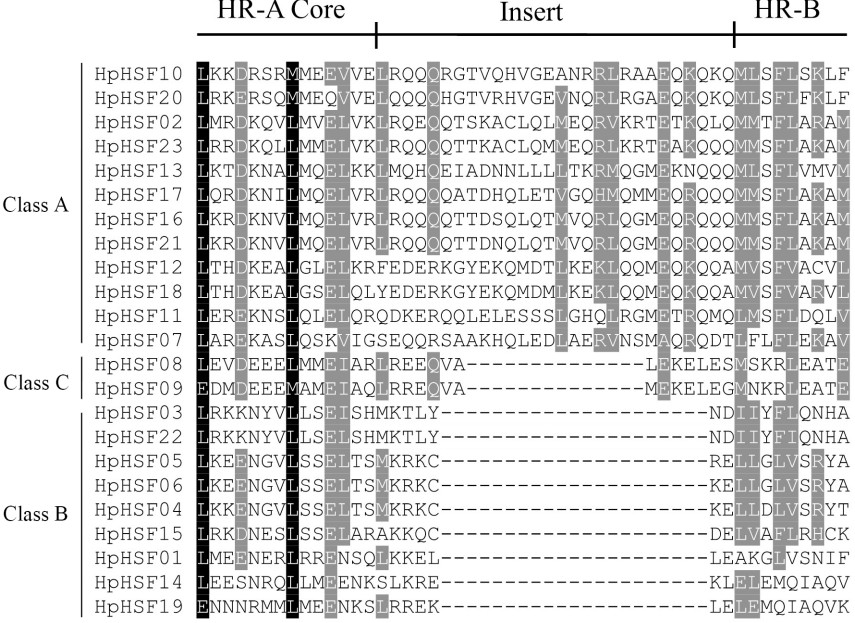

**Figure 2 Multiple sequence alignment of the HR-A/B regions of 23 members of the HSF protein family.** The annotations at the top describe the location and boundaries of the HR-A core, insert, and HR-B region within the HR-A/B region.

*HpHSF10*, *HpHSF20*); HpHSF B is further divided into 5 subgroups (B1-B5) consisting of nine members (*HpHSF01*, *HpHSF03*, *HpHSF04*, *HpHSF05*, *HpHSF06*, *HpHSF14*, *HpHSF15*, *HpHSF19*, *HpHSF22*), while HpHSFC only contained *HpHSF08* and *HpHSF09* in one subgroup. All of the HpHSFs in the phylogenetic tree were consistent with the classifications obtained from the HEATSTER database. HpHSF proteins were not clustered in A2, A7 and A9 sub-groups.
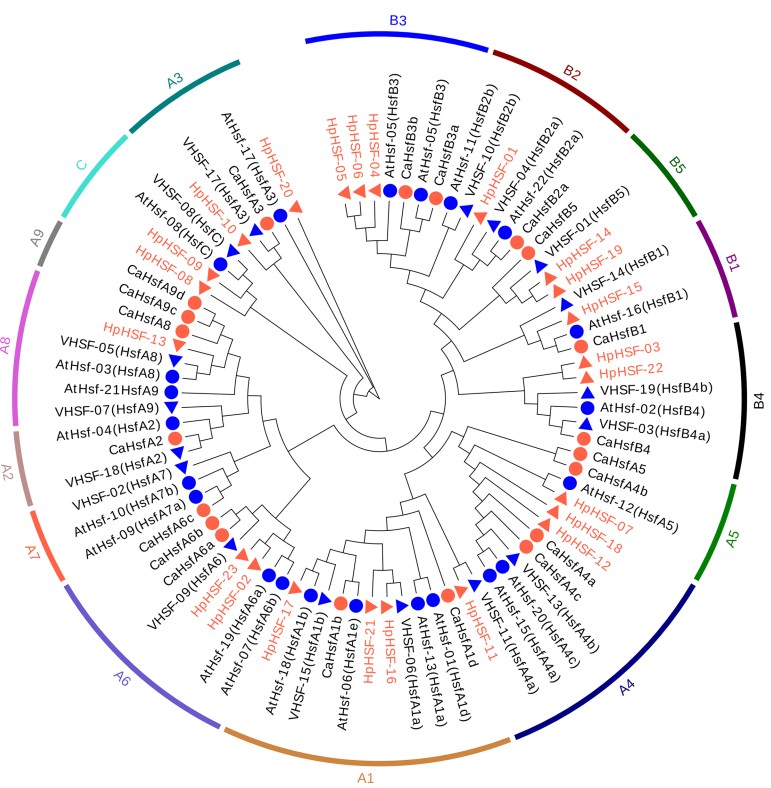

**Figure 3** **Phylogenetic tree of HSF proteins from *H. perforatum* (Hp), *Capsicum annuum L.* (Ca), *Vitis vinifera L.* (V), and *A. thaliana* (At).** The full-length of amino acid sequences of HSF proteins in the four species were used to construct the phylogenetic tree using MEGA 6, the statistical method used was the Neighbor-Joining (NJ) method, and the test of phylogeny was based on the bootstrap method with 1000 bootstrap replicates and pairwise deletion. The amino acid substitution model used was the p-distance model. Subclass numbers of *Arabidopsis*, pepper, and grape are listed.

## Gene structures analysis, conserved Motifs, and protein modeling of HpHSFs

The gene structures of *HpHSFs* were investigated through an analysis of the intron/exon boundaries, as shown in Fig. 4A. *HpHSF20* comprised three exons, and *HpHSF03* comprised four exons, except for the aforementioned two HpHSFs, all the other 21 HpHSFs contained two exons and one intron. The intron phases of HpHSFs were 0, except for phase 1 in *HpHSF20* and phase 2 in *HpHSF03*. In conclusion, the gene structure was conserved among the 23 HpHSF members.

In addition, the conserved motifs and phylogenetic relationships of all 23 HpHSF proteins were revealed via a systematic examination (Table 2 and Fig. 4B). The HSF domains DBD, OD (HR-A/B), NLS, RD (Repressor Domain), AHA, and NES were found in HpHSF protein sequences. Twelve, nine, and two HpHSF proteins were classified in subclasses A, B, and C, respectively. The HpHSF proteins in subclass A were characterized by DBD at the N-terminus followed by the HR-A/B motif. NLS, AHA, and NES were found in partial subclass A HSF proteins. The RD motif was only found in subclass B HpHSF sequences. Subclass C contained DBD, HR-A/B, and NLS, which were considered to be

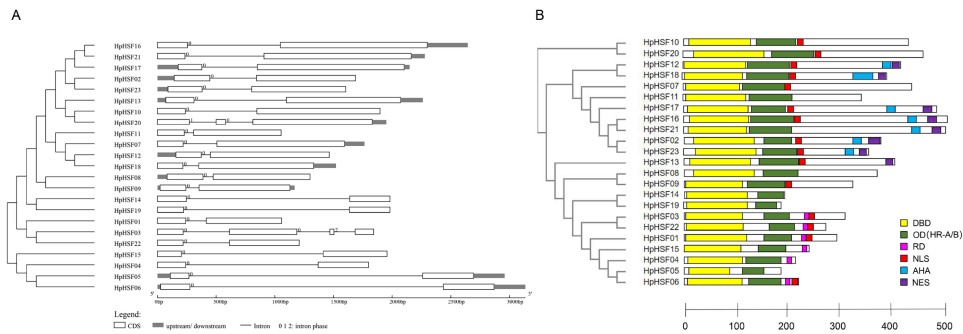

**Figure 4** **Gene structure (A) and conserved motifs (B) of HpHSF family members.** (A) Blank box, Grey box and black line were represented CDS, upstream/ downstream and intron, respectively. The number 0, 1, and 2 on the black line were intron phase. (B) DBD, OD (HR-A/B), RD, NLS, AHA and NES motifs of HpHSF members were identified by Heatster. The motifs were annotated and exhibited in different colored boxes.

highly conserved motifs in HpHSF proteins. The HpHSF proteins were modeled using the SWISS-MODEL program (Fig. 5). A *Drosophila* heat shock transcription factor was used as a template, and the template model was taken from the Protein Data Bank (SMTL ID: 1hkt.1). The HpHSFs shared approximately 40% sequence similarity and 30% query coverage. The start position of the α1 DBD domain was notated.

### *Cis*-acting elements analysis in the promoter regions of HpHSF genes

We searched for potential *cis*-acting elements in the 1.5 kb upstream sequences of the translation initiation codons of HpHSFs in the PlantCARE database, and the results revealed the presence of various *cis*-elements in the 5′ flanking regions associated with stress, hormones, and development (Ning et al., 2017). In stress-related *cis*-acting elements, some elements related to various stresses, such as light, low/high temperature, drought, anaerobic induction, and wounds were found in a large number of *HpHSF* genes, including heat-shock response elements (HSEs), TC-rich repeats, Myb-binding DNA sequences (MBSs), anaerobic induction elements (AREs), and low temperature range (LTR) (Fig. 6, Table S4). In addition, several hormone-related *cis*-acting elements were observed in the promoters: ABA-responsive elements (ABREs), MeJA responsive elements (TGACG-motif/CGTCA-motif), ethylene-responsive element (ERE), auxin-responsive elements (TGA-elements), and salicylic acid responsive elements (TCA-elements) were detected in the promoters of 19, 17, 13, 13 and, and seven *HpHSFs*, respectively. These findings suggested that the *HpHSF* genes might be involved in multiple transcriptional regulation mechanisms for plant growth and stress responses.

### Expression profiles of *HpHSFs* across different tissues

Based on the *H. perforatum* genes against RNA-seq data from four tissues—roots, stems, leaves, and flowers, a heat map of the transcription patterns of the HpHSF family was generated to explore the transcription patterns of *HpHSF* genes. RNA-seq data could be searched from the Sequence Read Archive (SRA-NCBI) with accession numbers SRR8438983 (flower), SRR8438984 (leaf), SRR8438985 (stem), and SRR8438986 (root).

**Table 2  Conserved domains and motifs of HpHSF proteins.**

| Gene name | Group | DBD | OD | NLS | NES | AHA | RD | RD+NLS |
|---|---|---|---|---|---|---|---|---|
| HpHSF01 | B2 | 12–130 | 159–209 | – | – | – | – | 238–265 |
| HpHSF02 | A6 | 34–136 | 141–223 | 224–235 | 360–378 | 329–342 | – | – |
| HpHSF03 | B4 | 9–109 | 174–211 | – | – | – | – | 244–267 |
| HpHSF04 | B3 | 14–114 | 129–185 | – | – | – | 194–208 | – |
| HpHSF05 | B3 | 19–89 | 112–159 | – | – | – | – | – |
| HpHSF06 | B3 | 17–117 | 125–181 | 209–224 | – | – | 190–204 | – |
| HpHSF07 | A5 | 11–107 | 125–197 | 197–215 | – | – | – | – |
| HpHSF08 | C | 38–136 | 156–220 | – | – | – | – | – |
| HpHSF09 | C | 9–107 | 128–192 | 197–212 | – | – | – | – |
| HpHSF10 | A3 | 17–120 | 137–205 | 209–227 | – | – | – | – |
| HpHSF11 | A4 | 10–123 | 136–216 | – | – | – | – | – |
| HpHSF12 | A4 | 6–119 | 122–202 | 205–225 | 392–407 | 374–390 | – | – |
| HpHSF13 | A8 | 15–116 | 142–213 | 213–222 | 381–395 | – | – | – |
| HpHSF14 | B5 | 10–128 | 149–190 | – | – | – | – | – |
| HpHSF15 | B1 | 1–101 | 144–193 | – | – | – | 226–237 | – |
| HpHSF16 | A1 | 23–119 | 129–222 | 223–241 | 480–495 | 433–451 | – | – |
| HpHSF17 | A1 | 5–101 | 102–195 | 198–216 | 462–477 | 412–430 | – | – |
| HpHSF18 | A4 | 6–119 | 122–202 | 205–225 | 380–395 | 322–378 | – | – |
| HpHSF19 | B5 | 9–120 | 142–183 | – | – | – | – | – |
| HpHSF20 | A3 | 29–158 | 174–242 | 246–264 | – | – | – | – |
| HpHSF21 | A1 | 16–112 | 122–215 | 215–233 | 473–488 | 426–444 | – | – |
| HpHSF22 | B4 | 9–109 | 177–224 | – | – | – | – | 247–270 |
| HpHSF23 | A6 | 30–132 | 137–219 | 220–242 | 342–360 | 312–325 | – | – |

**Notes.**
DBD, DNA-bind domain; OD, heptad repeat A (N-terminus) or B (C-terminus) domain; NLS, nuclear localization signal; NES, nuclear export signal; AHA, aromatic and hydrophobic amino acid residues embedded in an acidic context; RD, repressor domain.

According to the FPKM values, the expression profiles of the HpHSF gene differed considerably in the four samples (Fig. 7). Among class A members, *HpHSF12*, *HpHSF18*, and *HpHSF13* were expressed at high levels, while *HpHSF02* and *HpHSF23* were expressed at relatively low levels or were not detected. Moreover, the expression of *HpHSF11*, *HpHSF18*, *HpHSF13*, and *HpHSF07* in leaves was higher than that in other tissues. Among the class B families, *HpHSF15* was expressed at significantly high abundances in all tissues compared with other genes. *HpHSF03* and *HpHSF22* were expressed at low levels or not expressed at all. In general, members of the class B family were expressed at higher levels in the root than in other tissues, except *HpHSF01*, as well as the two members of class C, implying their critical roles in roots.

## Expression analysis of *HpHSF* genes under heat stress treatment

HSF genes were found to play an important role in plant thermotolerance. In our study, the expression patterns of the HpHSF gene family were determined using qRT-PCR to demonstrated how HSF genes respond to heat stress. As shown in Fig. 8, the expression of *HpHSF2, 12, and 21* did not significantly change with heat stress. *HpHSF03*, *11*, *18* and *22*

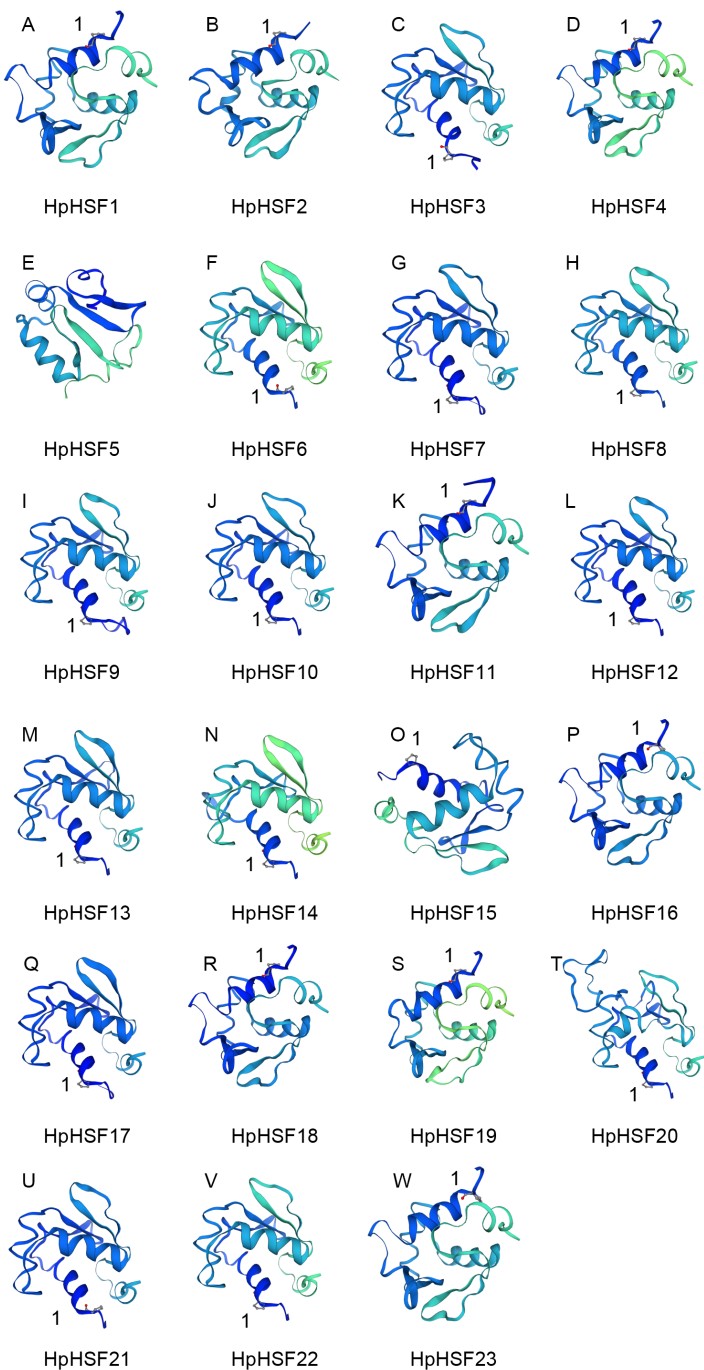

**Figure 5** (A–W) **Modelling of HpHSF family members, A *Drosophila* HSF was used as a template.** Templates corresponding to SMTL ID: 1hkt.1. The HpHSFs shared approximately 40% sequence similarity and 30% query coverage. The start position of the α1 DBD domain was notated by 1.

were repressed after heat stress treatment, and the remaining HpHSFs were up-regulated to varying degrees. Noticeably, the expression of *HpHSF10* increased dramatically, and was approximately 300 times higher at 3 h than the levels in the control, indicating that

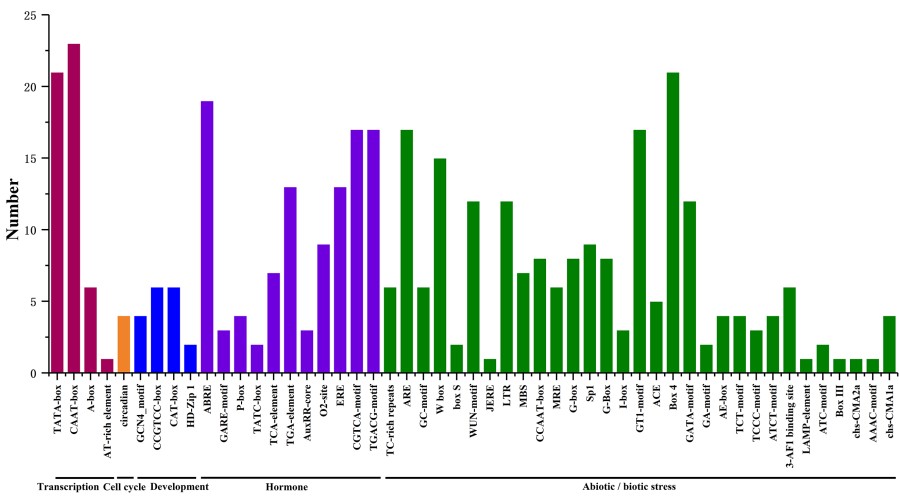

**Figure 6** **Number of *HpHSF* genes containing various *cis*-acting elements.** The graph was generated based on the presence of *cis*-acting elements responsive to specific processes/elicitors/conditions (*x*-axis) in *HSF* gene family members (*y*-axis).

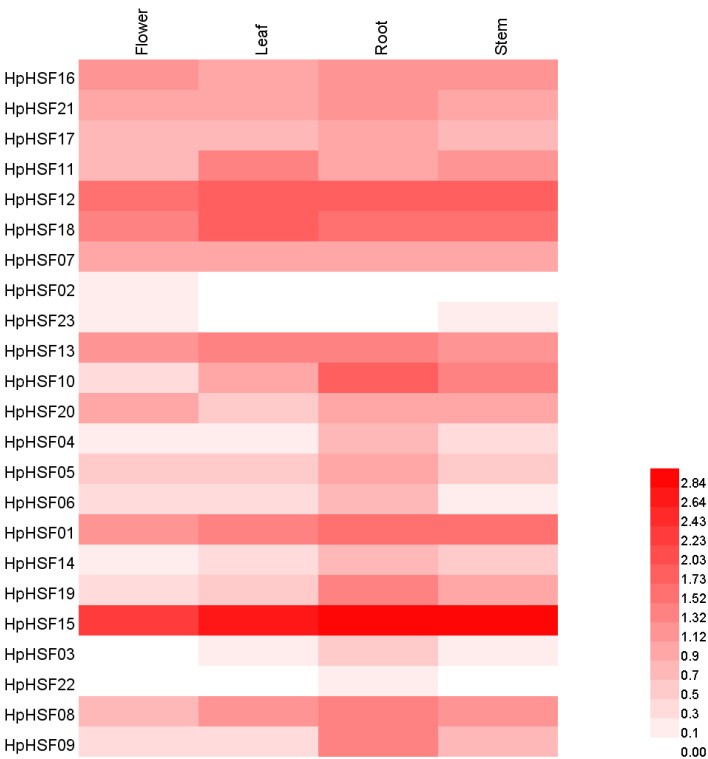

**Figure 7** **Heat map representation and hierarchical clustering of *HpHSF* genes in flower, leaf, root, stem.** The expression values were calculated by fragments per kilobase of exon model per million mapped (FPKM).

*HpHSF10* is involved in the pathway of heat stress response. In addition, the expression of *HpHSF1*, *14*, *20*, and *23* also changed considerably, and these genes are thus worthy of further consideration.

## DISCUSSION

Temperature is a key environmental factor affecting several physiological pathways in plants. Secondary metabolite production determines the immunologic defense and economic value of *H. perforatum*, which is a medicinal plant. The concentration of hypericin and pseudohypericin in *H. perforatum* is closely related to temperature. The heat tolerance and photosynthetic rates of *H. perforatum* are both significantly reduced at high temperatures, and the total hypericin content (hypericin + pseudohypericin) is lower following high temperature treatment (*Zobayed, Afreen & Kozai, 2005*). Heat stress has been demonstrated to be detrimental in other species, seed yield is reduced following exposure for five days to high temperatures in flax (*Gusta, O'Connor & Bhatty, 1997*). It is thus of great importance to study how medicinal plants respond to high temperature stress with regard to growth, metabolism, photosynthesis, and even global climate. The HSF gene family plays an important role in plant adaptations to various biotic or abiotic stresses, especially high temperature stress. HSFs regulate HSPs as a partner at the genetic and transcriptional level to improve high-temperature stress tolerance (*Wang et al., 2004*). In this study, the identification and characteristics of 23 *HSF* genes were investigated based on the *H. perforatum* genome database, and the expression profiles of these 23 genes were analyzed to explore their functions in heat stress response in *H. perforatum*. The number of HpHSFs was low in comparison with numbers identified in other species; the 23 non-redundant complete genes in *H. perforatum* were fewer than those in *G. raimondii* (57), *Salix purpurea* (48), and *Linum usitatissimum* (34). Overall, the isolation and identification of these *HSF* genes is helpful for illustrating the molecular genetic basis of *H. perforatum*. The expression patterns of *HpHSFs* in four tissues and response to heat stress at 42 °C suggested that the *HSF* gene family was ubiquitously expressed, and several *HpHSF* genes could play important roles in adaptation to environmental stress.

The essential structures and motifs of 23 HpHSF genes and amino acids were identified. The conversed motifs of HpHSF protein, DBD, OD (HR-A/B), NLS, NES, and RD, were annotated. The DBD domain consists of approximately 100 amino acid residues that are highly conserved in yeast, plants, and mammals (*Schultheiss et al., 1996*). Similar to the results of previous studies, our findings showed that many sequences are conserved based on phylogenetic relationships in *Arabidopsis*, pepper, grape, and *H. perforatum* and the coiled-coil structure of HR-A/B region prediction. The *HpHSF* genes were classified into three classes (A, B, and C). Classes A and B were further divided into 9 (A1-A9) and 5 (B1-B5) subclasses, respectively. The number of class A HSF genes varies in plants—15 in *Arabidopsis* and maize, 13 in rice and mungbean, and 16 in soybean. There were nine class B HSFs in *H. perforatum*. The number of class B HSFs identified in plants is 10 in mungbean, 8 in rice, 7 in maize, and 5 in *Arabidopsis*. Most of the subclasses are shared among several species but are not identical. The subclasses A2, A7, and A9 identified in our study have

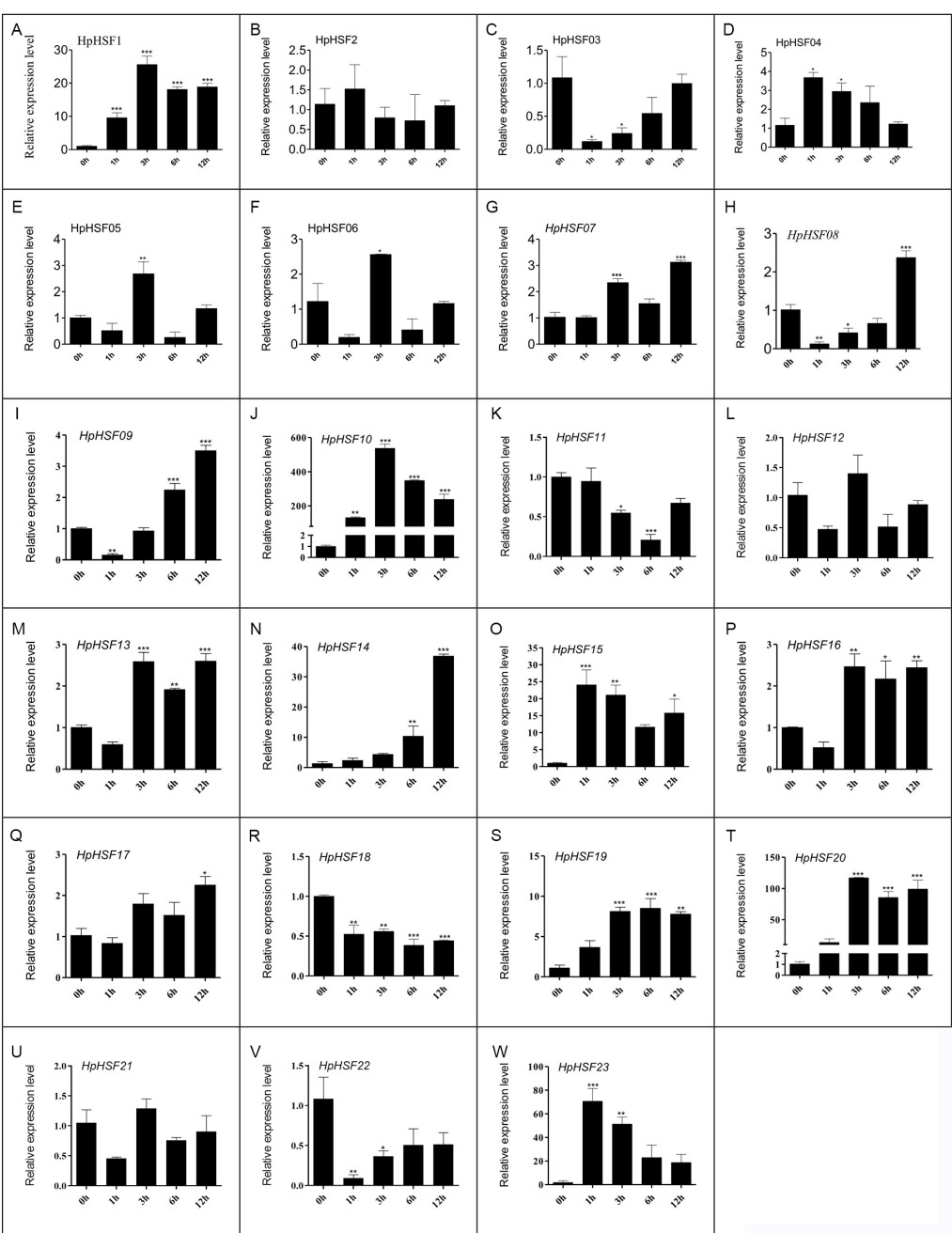

**Figure 8** **(A–W) Relative gene expression of *HpHSFs* analyzed by qRT-PCR response to heat stress treatment.** qRT-PCR data was normalized using *Hypericum perforatum Actin 2* gene and are shown relative to 0 h. X-axes are time course (0 h, 1 h, 3 h, 6 h and 12 h) and y-axes are scales of relative expression level. All Data represent means ± SD of three independent replicates. Statistical significance was analyzed by one way ANOVA (*, $P < 0.05$; **, $P < 0.01$; ***, $P < 0.001$).

been reported in some species, such as *Arabidopsis* and Arachis (*Wang et al., 2017*), but not in *H. perforatum*. It was hypothesized that elimination of introns, exon shuffling, and generations of exons might cause altered grouping in the phylogeny (*Nover et al., 2001*). Overall, these observations suggested the functional conservation and divergence of *HSF* genes among different plants.

HSF protein is involved in abiotic stress responses and hormone signaling in plants (*Huang et al., 2015*; *Zhang et al., 2015*). The *cis*-acting elements in the promoter region can regulate the transcription activity of the corresponding genes. Research on the detection of *cis*-acting elements could facilitate our understanding of the function and expression profiles of genes (*Fragkostefanakis et al., 2015*; *Wang et al., 2017*) . The promoter region of the *HpHSF* gene family members contains various elements related to growth and development, hormone responses, and stress responses. The numbers and types of elements vary among the *HpHSF* promoters, and overlapping phenomena existed in different genes, implying that the members of the family may regulate a variety of abiotic stresses and plant hormone signaling pathways simultaneously. This reflects the diversity and complexity of the biological functions of the *HpHSF* gene family.

Gene expression profiles in different tissues are usually closely correlated with their functions in organ development (*Guo et al., 2008*). In this study, the expression patterns of *HpHSF* genes in four different tissues were investigated. Remarkably, the expression of *HSF15* was found to be the highest among all genes in all four tissues. Each gene was expressed differently in the four tissues, such as *HSF10*, which had the highest expression in roots but the lowest expression in flowers, and *HSF18*, the expression of which was higher in leaves than in other tissues, indicating their potential function in roots and leaves, respectively. All these *HpHSF* genes play roles in different tissues to ensure the normal development of plants. The low expression in certain organs of some HSFs does not mean that they have no function in these organs. Tissue-specific expression patterns of identified *HpHSF* genes indicate that HpHSFs are widely involved in the growth and development of various tissues, indicating an important role for studying the functions of *HpHSF* genes in *H. perforatum* developmental biology.

Plant HSFs play a central role in eliciting the expression of genes encoding heat shock proteins (HSPs) or other stress-inducible genes (*Nishizawa-Yokoi et al., 2009*; *Scharf et al., 2012*), which are important for plant tolerance to heat or other stress conditions. According to previous reports, the genome-wide expression profile suggested that several *HSF* genes are transcribed at relatively high levels during heat stress (*Chung, Kim & Lee, 2013*; *Giorno et al., 2012*).

In this study, the 23 *HpHSF* genes with specific sequence features and amino acid motifs were identified. Base on the motifs, the *HpHSF* genes were phylogenetically divided into three broad groups. The abiotic stress-related *cis*-acting elements were identified in the promoter of *HpHSF* genes. The expression of 23 *HpHSF* genes in different tissues and distinct patterns during heat treatment was performed. Among these genes, 14 *HpHSF* genes were upregulated (>2-fold) and 4 (*HpHSF3, 11, 18, 22*) were downregulated during the heat stress treatment. Specifically, *HpHSF10* was the most strongly induced (~300-fold) in response to heat stress; *HSF20* expression was more than 90 folds that of the control

after heat treatment; the expression levels of *HSF14*, *HSF15*, and *HSF23* were about 20 times higher than those in the control group, indicating that these *HpHSF* genes were very sensitive with a strong heat stress response. These genes play an important role in regulating the response of *H. perforatum* to heat stress and warrant further attention and exploration. All the systematic and phylogenetically analysis of *HpHSF* genes contributed to the genomic improvement and medical values of *H. perforatum* for high temperature tolerance.

## CONCLUSIONS

In conclusion, a comprehensive analysis of the *HpHSF* gene family with regard to the genomic structures, conserved motifs, phyletic evolution, *cis*-acting elements, and expression patterns was performed in this work. Overall, the bioinformatic analyses and expression profile studies of HSFs are helpful in understanding the important role of HSFs in *H. perforatum*'s response to heat stress and providing the foundation for exploring methods to understand and regulate these stress responses.

## ACKNOWLEDGEMENTS

I would like to express my gratitude to all the people who helped me with writing this article. Finally, I am indebted to my beloved family for their consistent support and encouragement.

### Funding

This research was supported by the National Natural Science Foundation of China (31670299, 31870276, 31900254, 31800259) and the Project of the National Key Technologies R & D Program for Modernization of Traditional Chinese Medicine (2017YFC1701300, 2019YFC1712602), Fundamental Research Funds for the Central Universities (GK202003056). The funders had no role in study design, data collection and analysis, decision to publish, or preparation of the manuscript.

### Grant Disclosures

The following grant information was disclosed by the authors:
National Natural Science Foundation of China: 31670299, 31870276, 31900254, 31800259.
Modernization of Traditional Chinese Medicine: 2017YFC1701300, 2019YFC1712602.
Fundamental Research Funds for the Central Universities: GK202003056.

### Competing Interests

The authors declare there are no competing interests.

### Author Contributions

- Li Zhou conceived and designed the experiments, performed the experiments, analyzed the data, prepared figures and/or tables, authored or reviewed drafts of the paper, and approved the final draft.

- Xiaoding Yu and Wen Zhou conceived and designed the experiments, prepared figures and/or tables, and approved the final draft.
- Donghao Wang conceived and designed the experiments, authored or reviewed drafts of the paper, and approved the final draft.
- Lin Li performed the experiments, analyzed the data, prepared figures and/or tables, authored or reviewed drafts of the paper, and approved the final draft.
- Qian Zhang conceived and designed the experiments, performed the experiments, prepared figures and/or tables, and approved the final draft.
- Xinrui Wang and Sumin Ye analyzed the data, authored or reviewed drafts of the paper, and approved the final draft.
- Zhezhi Wang conceived and designed the experiments, analyzed the data, authored or reviewed drafts of the paper, and approved the final draft.

## Data Availability

The raw measurements are available in the Supplementary Table.

## Supplemental Information

Supplemental information for this article can be found online at http://dx.doi.org/10.7717/peerj.11345#supplemental-information.

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
