# Peer review of "Genome-wide identification, classification and expression profile analysis of the HSF gene family in Hypericum perforatum"

_PeerJ, doi:10.7717/peerj.11345_

## Round 0.1 · original submission · Major Revisions

Please address the critiques of both reviewers and revise your manuscript accordingly.

·

Basic reporting

The Ms entitled ' Genome-wide identification, classification and expression profile analysis of the HSF gene family in Hypericum perforatum (#52717) by Zhou et al. has been critically assessed all for the analytical details, finding of scientific novelty, and linguistic representations. Honestly, genetic factors of high-temperature stress in plant systems are of scientific interest keeping in view the rising concerns of global warming and its effect on agriculture and the ecosystem. Therefore, the appropriateness of the article may be considered. However, the authors could have made a better effort to bring significant findings from their analysis. Several issues in the article require serious attention before can be considered for publication in a prestigious journal like PeerJ.

Overall, the Manuscript needs a thorough check for clarity of sentences, rechecking of all typos, grammar, and usage of the English language. Several sentences are unclear due to ambiguous statements and may need recasting them to improve the clarity of statements. On several occasions, authors have left without the citation year in text, which indicates their very casual manuscript finalization before the submission. My strong recommendation is to get the manuscript checked from any Native English speaking person or take the help of a professional language editing service.

The 'Introduction section' may be highlighted with the need to analyze HSFs in the plant system with prior work or citations. Several minor suggestions were mentioned in the pdf article itself. Some of the References cited are old and maybe replaced with recent citations. May include article: Sci Rep. 9, 5581 (2019). https://doi.org/10.1038/s41598-019-41936-1 and cite accordingly.

Experimental design

In the M&M section, authors may use the 'Heatster' program to annotate their sequences and compare them with the Phylogenetic groupings. Also, authors may need to include more in silico analysis like the evolutionary significance of HSF gene expansion and protein modeling with active sites.

Authors may clearly mention the Phylogenetic analysis parameters, like the Alignment method, amino acid substitution Models used, basis of selecting the models, etc. Consider using MEGA-X in MUSCLE. May consult https://doi.org/10.1038/s41598-019-41936-1 for details.

Validity of the findings

The results section may be strengthened further with the suggested new analysis.

In the Discussion section, the authors fail to sufficiently explain any scientific contribution to the field. The authors may need to revise this section to bring out the salient and new findings from their study.

Similarly, authors may also strengthen the 'Concluding section' with highlights of their study and how it is beneficial or in what way it may contribute to the scientific prosperity of the plant per se.

The authors may check and add references as suggested for the manuscript.

Additional comments

Authors may recheck the Ms again and again to minimize all typographic and grammatical errors. Also, they may need to bring their best efforts to reveal any significant scientific findings from their study. With these major revisions, I am sure their manuscript will be improved to be considered for publication in the PeerJ journal.

Reviewer 2 ·

Basic reporting

In this manuscript the authors identified and characterized several heat shock transcription factors in H.perforatum using various bioinformatic analysis tools. Considering the important role of HSFs in a plant’s response to heat stress, it is important to perform bioinformatic analyses and expression profile studies of HSFs to better understand this response pathway. Given the importance of H.Perforatum extracts in pharmacological applications and the challenges in its production, it is important to identify HSF genes in this plant species which could provide the foundation for exploring methods to understand and regulate these stress responses. Studies such as the work in this manuscript contribute towards such efforts. This work is within the scope of PeerJ journal. The manuscript is written in professional English for the most part and the structure conforms to PeerJ standard. Most figures are of publication quality and appropriately annotated. The raw data relevant to the manuscript is available. A few minor points are noted below.
1. Figure 1: HpHSF23 is missing in the figure.
2. Figure 3: HpHSF08 is listed as a member of group A. However, according to the figure it should be HpHSF18
3. Figure 3: A3 is missing while A9 is duplicated. Is one of the A9 duplicates actually A3? Please correct the group names.
4. Reference figure 7 in paragraph on lines 218-229
5. Figure 8: Mention what the * indicates in figure legend
6. There are some minor grammatical errors in the manuscript such as line 41, 47,67,80, 114,173, 176 and 239 which upon correction will improve the quality of the manuscript.

Experimental design

The authors have defined the objective of the study well. The bioinformatic analyses and biochemical characterization are well structured and seem to be performed accurately following standard procedures in the field. However, the authors must clarify the following point to improve the overall clarity and transparency of the manuscript.
1. Please elaborate on the statistical analyses performed for the RT-qPCR assay to distinguish between significant and insignificant changes in the HSF expression levels.

Validity of the findings

Overall, the results are detailed and presented in a clear manner for the most part. Appropriate analyses have been performed and reported in accordance with standards in the field. The discussion of results is thorough, and the conclusion answers the original research question. There are some points outlined below which must be revised prior to acceptance of the manuscript.
1. Line 31: Please elaborate what “vegetarian attack” indicates.
2. Line 163: Was the finding that all the HpHSF proteins were unstable, an expected result? What does this finding imply in terms of the function of the protein?
3. Line 180: Please use the scientific names of the plants as that is what is used in the figure 3 title and mention the common names in parenthesis.
4. Line 191: Please clarify what “Peu” stands for?
5. Line 224-225: The text states “Moreover, the expression of HpHSF11, HpHSF18, HpHSF13 and HpHSF07 in leaf were higher than that in other tissues”. However, according to Figure 7, the expression of HpHSF12 appears to be higher than both HSF7 and HSF11. Is there a reason HpHSF12 was not included in the statement above?
6. Lines 226-228: HpHSF03 and HpHSF22 are expressed at very low levels or not expressed at all. Please comment on this in the text.
7. Lines 233-234: “As shown in Figure 8, the expression of HpHSF2, 11, 12, 21 had no significant change”. However, in figure 8, there are * on the bars and the levels do look different. Please clarify what the * denotes which would resolve this ambiguity.
8. Figure 8: For HpHSF 21 and 22 plots, why is the 0h time-point not at 1 on the Y-axis considering this is relative expression? If this is in error, please rectify.
9. Please clarify what plant tissues were used in the heat stress response experiments. If it was the entire seedling, please mention that.
10. Have the authors compared how the expression levels of HSFs change in response to heat stress in mature plants versus young plants/seedlings? If not, the authors should perform a comparative study to answer understand if it’s the same or different HSFs that are affected in young and mature plants. This might help to identify how similar or different the heat stress signatures are in the plant depending on its stage in the life cycle.

---

## Round 0.2 · accepted · Accept

Both reviewers are satisfied by the revision and the amended manuscript is acceptable now.

ssjm

·

Basic reporting

The revised version of the article by Zhou et al. has been carefully checked now. The improvements in the article are clearly visible compared to the previous version. The English language usage is clearer and references are made up to date now. The article may be accepted for publication.

Experimental design

The experimental designs suggested were mostly included and this fits well with the Aims and scope of the PeerJ. The research questions are now better stated to initiate a platform to work further in this direction.

Validity of the findings

The findings are now being discussed sufficiently towards acceptance.

Additional comments

Suggestions were mostly addressed by the authors. no new suggestions are made.

Reviewer 2 ·

Basic reporting

The revised manuscript and the rebuttal letter adequately address the previously raised issues.

Experimental design

The revised manuscript and the rebuttal letter adequately address the previously raised issues.

Validity of the findings

The revised manuscript and the rebuttal letter adequately address the previously raised issues.

Additional comments

The revised manuscript and the rebuttal letter adequately address the previously raised issues.